# Zebrafish as a Screening Model to Study the Single and Joint Effects of Antibiotics [note 1]

**DOI:** 10.3390/ph14060578

**Published:** 2021-06-17

**Authors:** Roxana Jijie, Gabriela Mihalache, Ioana-Miruna Balmus, Stefan-Adrian Strungaru, Emanuel Stefan Baltag, Alin Ciobica, Mircea Nicoara, Caterina Faggio

**Affiliations:** 1Marine Biological Station “Prof. dr. I. Borcea”, “Alexandru Ioan Cuza” University of Iasi, Nicolae Titulescu Street, No. 163, 9007018 Agigea, Romania; baltag.emanuel@gmail.com; 2Department of Exact and Natural Sciences, Institute of Interdisciplinary Research, “Alexandru Ioan Cuza” University of Iasi, 11 Carol I, 700506 Iasi, Romania; balmus.ioanamiruna@yahoo.com (I.-M.B.); strungaru_stefan@yahoo.com (S.-A.S.); 3Integrated Center of Environmental Science Studies in the North Eastern Region (CERNESIM), “Alexandru Ioan Cuza” University of Iasi, 11 Carol I, 700506 Iasi, Romania; gabriela.mihalache@uaic.ro; 4Department of Horticultural Technologies, “Ion Ionescu de la Brad” University of Agricultural Sciences and Veterinary Medicine, 700440 Iasi, Romania; 5Department of Biology, Faculty of Biology, “Alexandru Ioan Cuza” University of Iasi, B-dul Carol I, 700505 Iasi, Romania; alin.ciobica@uaic.ro (A.C.); mirmag@uaic.ro (M.N.); 6Doctoral School of Geosciences, Faculty of Geography-Geology, “Alexandru Ioan Cuza” University of Iasi, B-dul Carol I, 700505 Iasi, Romania; 7Department of Chemical, Biological, Pharmaceutical and Environmental Sciences, University of Messina, Viale F. Stagno, d’Alcontres, 31 98166 S. Agata-Messina, Italy

**Keywords:** zebrafish, antibiotics, aquatic environment, toxicity, combined effects

## Abstract

The overuse of antibiotics combined with the limitation of wastewater facilities has resulted in drug residue accumulation in the natural environment. Thus, in recent years, the presence of antibiotic residues in the environment has raised concerns over the potential harmful effects on ecosystems and human health. The in vivo studies represent an essential step to study the potential impact induced by pharmaceutical exposure. Due to the limitations of traditional vertebrate model systems, zebrafish (*Danio rerio*) has recently emerged as a promising animal model to study the toxic effects of drugs and their therapeutic efficacy. The present review summarizes the recent advances made on the toxicity of seven representative classes of antibiotics, namely aminoglycosides, β-lactams, macrolides, quinolones, sulfonamides, tetracyclines and polyether antibiotics, in zebrafish, as well as the combined effects of antibiotic mixtures, to date. Despite a significant amount of the literature describing the impact of single antibiotic exposure, little information exists on the effects of antibiotic mixtures using zebrafish as an animal model. Most of the research papers on this topic have focused on antibiotic toxicity in zebrafish across different developmental stages rather than on their efficacy assessment.

## 1. Introduction

Antibiotics are natural, synthetic or semi-synthetic compounds that are able to kill or inhibit the growth and metabolic activity of microorganisms. Besides treating or preventing infectious diseases, antibiotics are often used as growth promoters [1,2]. Based on chemical and molecular structure, the most common classes of antibiotics are aminoglycosides, β-lactams, macrolides, quinolones, sulfonamides, tetracyclines and polyether antibiotics [3]. Moreover, the antibiotics can be divided into two groups based on their mechanism of action, which are either bactericidal (kill the bacteria) or bacteriostatic (inhibiting bacterial growth). As a result of the extensive use of antibiotics, combined with poor absorption and high-water solubility, it was just a matter of time until they accumulated in soil and water bodies and posed environmental and health threats. Although the half-life time of most antibiotics is between a few hours and several days, the residuals are considered to be a “pseudo-persistent” organic contaminant due to the continuous release into aquatic environments [4,5]. There are still no regulations concerning the levels of antibiotics in the surface water. The European Commission has recently included amoxicillin, ciprofloxacin, erythromycin, clarithromycin and azithromycin on the updated Watch List under the Water Framework Directive. According to the literature, most of the antibiotics that are found in aquatic environments are toxic to aquatic organisms [2].

In vivo studies represent an important step in drug testing effects. The limitations of traditional mammalian models motivated the researchers to search for alternative vertebrate model systems. In this context, the use of zebrafish (*Danio rerio*) to study the toxic effects of molecules or therapeutic efficacy has grown markedly in recent decades [6,7,8,9,10,11]. Their rapid life cycle, small size, external development, prolific spawning and genetic and physiological similarities with mammals are some of the main attractions of zebrafish as an alternative to traditional organism models for toxicology studies. The zebrafish genome has been fully sequenced, and a comparison with humans reveals that 70% of human genes have a zebrafish homologue [12], providing a useful resource for high-throughput analyses. In addition, because the embryos and early adults are optically transparent, a direct observation of internal organs and morphological characteristics by light or fluorescence microscopy is possible [13,14,15,16]. Furthermore, the facilities required to house zebrafish for scientific purposes are relatively simple and cheaper, and the experiments are performed across different developmental stages (e.g., embryo, larva or adult) and in a wide variety of laboratory conditions (e.g., large number of compounds from diverse classes in parallel, acute and chronic testing and a wide range of concentrations of the chemicals tested) [17,18,19,20,21]. Generally, the zebrafish acute and chronic toxicity tests are performed according to the guideline 203 of OECD (updated 2019) and guideline 230 of OECD (2009) [22,23,24,25,26], while guideline 236 of OECD (2013) is used to determine the acute toxicity of chemicals on embryonic stages [27,28,29].

Most of the research papers on this topic have reported that the zebrafish health or/and its capacity to respond to environmental changes are affected following both acute and chronic exposure to antibiotics. The toxicological effects of single and mixed antibiotics in zebrafish were studied by proteomic [30] and transcriptome sequencing [31,32], examination of behavioral activities [28,33,34] and biomarkers [35,36], histopathological analysis of various tissues [30,37], fish lethality assays (LC_50_) [22,38,39] and following the fish morphological changes [40]. Antibiotics were found to interfere with the bacterial communities of zebrafish gut and with fish development, causing malformations, such as hatching delay, shorter body length, yolk sac edema, curved body axis and pericardial edema [23,25,41,42,43,44]. Moreover, exposure to antibiotics was observed to impact the survival, behavior alterations (impacts in locomotion, learning, memory, exploration and aggression) and adverse effects on swimming performance, as well as changes in the social interaction behavior [22], of zebrafish. Due to the integrated behavior responses to internal (e.g., physiological and/or biochemical) and external (e.g., environment, social) indicators, adverse signs induced by antibiotic exposure can be assessed early. It has also been found that antibiotics can elicit cardiotoxicity, genotoxicity, nephrotoxicity, ototoxicity, hepatotoxicity, reproductive toxicity, immunotoxicity and oxidative damage in zebrafish, without affecting fish survival (Table 1 and Table 4). Recently, the partial recovery capacity of the zebrafish after antibiotic exposure has been demonstrated [43]. 

It is very interesting to note that the concentrations of antibiotics used in most of the research papers on this topic are based on the previously reported concentrations found in the water bodies (e.g., surface waters and wastewaters), generally ranging from μg/L to mg/L levels. For example, a monitoring study has shown that sulfonamides have been frequently detected in surface waters and wastewaters with concentrations ranging from 8.4 ng/L to 211 μg/L [45]. Erythromycin concentrations in surface water bodies were found ranging from 0.1 ng/L to 1 μg/L [29]. In Tai Lake (China), the maximum concentrations increased in the order of macrolides (48.8 ng/L) < quinolones (210.67 ng/L) < β-lactams (361.74 ng/L) < tetracyclines (551.18 ng/L) < amphenicols (2.7 μg/L) < sulfonamides (10.2 μg/L) [46]. Additionally, high levels of norfloxacin (up to 2.8 μg/L), ofloxacin (up to 2.4 μg/L) and azithromycin (up to 1.7 μg/L) have been reported in the influent of 14 municipal wastewater treatment plants [47]. As expected, hospitals were a significant source of antibiotics, as sulfamethoxazole (0.4–2.1 μg/L), trimethoprim (2.9–5 μg/L), ofloxacin (25.5–35.5 μg/L), ciprofloxacin (0.85–2 μg/L), lincomycin (0.3–2 μg/L) and penicillin G (0.85–5.2 μg/L) were measured in hospital effluents in New Mexico [48]. Another study performed in Coimbra (Portugal) reported high concentrations of ofloxacin (0.3–10.6 μg/L) and ciprofloxacin (0.1–11 μg/L) in hospital effluents [49]. The concentration of enrofloxacin in the swine wastewater was up to 1.793 mg/L and levels of >0.1 mg/L were observed for sulfamonomethoxine, trimethoprim, ciprofloxacin, ofloxacin, lincomycin and tetracycline from certain pig farms [50]. The removal percentage of oxytetracycline in the wastewater treatment was 38 ± 10.5%, and the antibiotic concentration was still as high as 19.5 ± 2.9 mg/L in the treated outflow [51]. In addition, the concentration slightly decreased along the river to 377 ± 142 μg/L at a site ~20 km away from the discharging point. The maximum concentrations of tetracycline, oxytetracycline and doxytetracycline were found in the dry season as 11.16, 18.98 and 56.09 ng/L, respectively, through a survey of water sources along the lower Yangtze River (China) [52]. In summary, the concentration and distribution of antibiotics in aquatic environments are influenced by climatic properties, as well by the pollution source and the physico-chemical characteristics of antibiotics. 

Moreover, the drug toxicity or efficacy is tested by direct administration into the zebrafish water, with daily renewing of exposure solutions. Thus, the antibiotic molecules are absorbed through the fish skin, gills and gut, making it difficult to quantify the amount of drug uptake by each zebrafish. Few studies performed a quantitative determination and assessed the toxicity using a defined volume of antibiotics. For example, defined volumes of gentamicin were injected into the cardiac venous sinus of larval zebrafish [53]. Zhang et al. [54] assessed the toxicity and performed a quantitative determination of four antibiotics with different polarities in zebrafish by using a liquid chromatography–tandem mass spectrometry (LC-MS/MS) method in multiple reaction monitoring. To overcome the above shortcoming, analytical techniques for simple, rapid and sensitive measurements of antibiotic concentration in fish need to be developed. It is well known that antibiotics can degrade in the environment under the effects of temperature, pH or light. In other words, the drug solubility, stability, concentration and administration, as well as the adsorption, distribution, metabolism and elimination of the antibiotics remain a concern.

The present paper outlines the recent advances made on the toxicological impact of seven representative antibiotic classes, namely aminoglycosides, β-lactams, macrolides, quinolones, sulfonamides, tetracyclines and polyether antibiotics, on zebrafish across different developmental stages (e.g., embryo, larva and adult), as well as the joint effects of antibiotic mixtures. Taken together, the results highlight that there are serious concerns about the side effects induced by antibiotic exposure in fish, and further research and strategies to prevent them from reaching the environment need to be developed.

## 2. Effects of Individual Antibiotics in Zebrafish

In recent years, considerable efforts have been devoted to the occurrence, fate and impact of biologically active compounds in the aquatic environment. According to their biological activity, these compounds can be classified in antibiotics, analgesics, anti-histamines, anti-inflammatories, etc. [40] Amongst the pharmaceuticals, the antibiotics have been broadly studied. It has to be highlighted that the discovery of antibiotics and their widespread use revolutionized human health after the Second World War. Before the availability of antibiotics, there were no efficient treatments for infectious diseases. Apart from their therapeutic usage, they can be used for prophylactic purposes or as growth promoters in animal breeding [55]. Antibiotics can reach the aquatic environment through different ways, such as the discharging of contaminated residues or the discarding of unused drugs. Some of them can be easily degraded, as in the case of penicillin, but some (fluoroquinolones and tetracyclines) last for longer periods of time, accumulating in concentrations that can exceed 1 mg/L [56]. The main concerns related to the use and presence of antibiotics in the environment are anti-microbial resistance and the toxicity to organisms. The results revealed that antibiotics may have a negative impact on organisms in the aquatic ecosystems, including freshwater algae, microphytes, macrophytes, zooplankton and fishes [57,58,59,60,61]. Commonly used fish for toxicity testing are rainbow trout (*Oncorhynchus mykiss*) [62,63], common carp (*Cyprinus carpio*) [45,64,65], goldfish (*Carassius auratus*) [66], medaka (*Oryzias latipes*) [60,61,67,68,69], gilthead seabream (*Sparus aurata*) [70,71,72], Tra catfish (*Pangasianodon hypophthalmus*) [73] or zebrafish (*Danio rerio*). For example, acute (0.01–10 mg/L) and chronic exposure (0.05–0.8 μg/L) to erythromycin caused histological changes in the gills and liver of rainbow trout [62]. In addition, oxidative effects and genotoxicity have been reported following the exposure of rainbow trout to oxytetracycline [63]. Many changes were found in the hematological profile and biochemical parameters of *Cyprinus carpio* upon chronic exposure to sulfamethoxazole (25–200 μg/L) [65]. The activity of brain acetylcholine esterase in male goldfish was significantly inhibited by norfloxacin (≥0.4 mg/L) and the mixture of sulfamethoxazole and norfloxacin (≥0.24 mg/L) after 4 days of exposure [66]. Moreover, the activities of 7-ethoxyresorufin O-deethylase, glutathione S-transferase and superoxide dismutase in liver and serum vitellogenin were increased by norfloxacin and mixtures. Treatment with various doses of erythromycin ranging from 2 μg/L to 2 mg/L for 96 h decreased the burst speed and swimming speed in medaka [68]. A 10 day treatment with sulfadiazine and trimethoprim in gilthead seabream juveniles induced significant changes in the expression of 41 liver proteins [74]. Additionally, enrofloxacin has been demonstrated to induce lipid peroxidation and neural dysfunction in Tra catfish [73]. In recent years, many studies have focused on the toxic effects of single antibiotics on embryos, larvae or adult zebrafish. Developmental toxicity, oxidative stress, nephrotoxicity, cardiotoxicity, oculotoxicity, neurotoxicity or ototoxicity (Table 1) have been observed following both acute and chronic exposure to antibiotics. 

**Table 1 pharmaceuticals-14-00578-t001:** Overview of the potential toxicological effects of individual antibiotics on zebrafish, dating from 2005 to the present.

Class.	Antibiotics	Concentrations (Weight/Water Volume)	Exposure Period	Lifespan Stages	Effects/Toxicity Observed	Refs.
Aminoglycosides	Netilmicin	10–1000 ng/mL	3 days at 6–72 hpf and at 3–6 dpf	Embryos at 6 hpf and larvae at 3 dpf	- Cardiotoxicity - Mild teratogenicity	[54]
	Gentamicin	Injected with defined volumes of 10 mg/mL	From 55 to 72 and 96 hpf	Larvae	- Nephrotoxicity	[53]
		0.2, 1, 2 and 5 μM	24 h	5 dpf	- Ototoxicity	[75]
		5, 10 and 20 μM	24 h	5 dpf	- Behavioral changes	[33]
		10 μM	48 h	3 dpf	- Oculotoxicity	[76]
		0.1–400 μM	Acute and chronic exposure	5–6 dpf	- Ototoxicity	[77]
	Neomycin	0.16, 1.6, 8 and 16 μM	24 h	5 dpf	- Ototoxicity	[75]
		125 μM	1 h	5 dpf	- Behavioral changes	[33]
		0.1–400 μM	Acute and chronic exposure	5–6 dpf	- Ototoxicity	[77]
	Streptomycin	0.1–400 μM	Acute and chronic exposure	5–6 dpf	- Ototoxicity	[77]
		0.1, 1 and 10 μg/mL	10 days	5 dpf	- Dysbiosis - Early mortality	[78]
	Etimicin	Survival test: 2, 5, 10, 20,50, 100, 200 and 500 mg/L	At 24, 48, 72, 96 and 120 hpf	6 hpf	- Low nephrotoxicity and ototoxicity compared with amikacin and gentamicin	[79]
	Amikacin	Hatching and toxicity: 750, 1500 or 2000 mg/L	From 1 to 3 dpf	1 dpf	- More developmental toxicities to the embryos than gentamicin	
β-Lactams	Amoxicillin	100 mg/L	7 days	Young zebrafish	- Behavioral changes - Oxidative stress	[24]
		0, 75, 128, 221, 380, 654 and 1125 mg/L	96 h	Embryos	- Developmental toxicity - Oxidative stress	[23]
		0, 1, 10, 25, 50 and 100 mg/L		Adults		
	Cefotaxime	10–1000 ng/mL	3 days at 6–72 hpf and at 3–6 dpf	Embryos at 6 hpf and larvae at 3 dpf	- Cardiotoxicity - Teratogenicity increased in a dose-dependent manner	[54]
		0, 10, 20 and 30 mM	1 day	5 dpf	- Locomotor toxicity - Abnormal expression of different genes	[80]
	Ceftazidime	0, 6.25, 12.5, 25 mg/L	96 h	Adult	- Behavioral changes	[34]
		10–1000 ng/mL	3 days at 6–72 hpf and at 3–6 dpf	Embryos at 6 hpf and larvae at 3 dpf	- Cardiotoxicity	[54]
Macrolides	Erythromycin, clarithromycin, azithromycin, midecamycin, josamycin	0, 0.1, 1, 10, 100 and 1000 μM	48 h until 5 dpf	3 dpf	- Hepatotoxicity	[81]
		0, 0.5, 1, 2	6 dpf	3 dpf	- Hepatotoxicity	[82]
	acetylspiramycin	0, 0.25, 0.5, 1	6 dpf	3 dpf	- Hepatotoxicity	[82]
	Erythromycin	0, 0.017, 0.034, 0.068, 0.136, 0.272 mM	96 h	24 hpf	- Cardiotoxicity	[27]
		0, 2, 20, 200 and 2000 μg/L	96 h	adult	- Abnormal gene expression - Behavioral changes	[68]
		0, 0.001, 0.01, 0.1, 1 and 10 μg/L	96 hpf	2 hpf	- Cardiotoxicity - Developmental toxicity - Enhanced swimming activity	[29]
	Tilmicosin	0, 0.63, 1.25, 2.5, 5, 10, 20 and 40 mg/L	4 dpf	2 dpf	- Developmental toxicity - Cardiotoxicity - Teratogenic effects - Oxidative stress - Apoptosis in embryos	[36]
Quinolones	Ciprofloxacin	0, 6.25, 12.5, 25 mg/L	96 h	Adult	- Behavioral changes	[34]
		5 μg/L	6–9 dpf	Larvae	- Increased expression of immune system cytokine genes	[83]
	Norfloxacin	0, 2, 20 and 200 μg/L	3 weeks	Larvae	- Reproductive toxicity	[84]
		600, 900, 1200 mg/L	72 hpf	Embryos	- Neurotoxicity - Impaired hatching rate - Increased mortality - Malformation	[85]
	Levofloxacin, gatifloxacin, norfloxacin, sparfloxacin, gemifloxacin, enoxacin, pefloxacin, prulifloxacin, lomefloxacin, moxifloxacin, ciprofloxacin, antofloxacin, 2-methylpiperazine, n-methylpiperazine, 4-n-Boc-2-methyl-piperazine	0, 0.1, 0.2, 0.5, 1, 2, 5, 10, 20 mM	6–72 hpf	Embryos	- Head deformation - Shortened tail - Tail tip malformation - Scoliosis and spinal curvature—reduced pigmentation - Developmental retardation - Pericardial edema - Heart malformation - Death	[86]
	Enrofloxacin	0.01, 1, 100 μg/L	120 h	Larvae	- Decreased body lengths - Deformed body shape - Disrupted metabolic processes	[87]
	Ofloxacin, Enrofloxacin	5 μg/L	20 days	Adult	- Accumulation in liver, skin, muscles and gills	[88]
	Gatifloxacin, Ciprofloxacin	- 413, 1238, 3713, 4239 mg/L - 156, 469, 1407 and 1949 mg/L	24 h	Adult	- Cardiotoxicity	[89]
Sulfonamides	Sulfamethoxazole	0, 0.09, 0.19, 0.39, 0.79, 1.58 mM	96 h	24 hpf	- Developmental toxicity - Cardiotoxicity (pericardial edema, bradycardia) - Oxidative stress	[27]
		80 and 100 mg/kg body weight per day	6 weeks	Adult	- Higher digestive enzyme activities	[41]
		0, 50, 100 and 500 μg/L	14 days	Adult	- Lipid peroxidation	[90]
		260 ng/L	6 weeks	Adult	- Impair the gut health - Higher metabolic rate	[91]
		0, 2, 20 and 200 μg/L	3 weeks	Larvae	- Developmental toxicity - Reproductive toxicity - Oxidative stress - Impact on the development of the zebrafish offspring	[84]
		0, 0.1, 1, 10 and 100 μg/L	120 h	5 hpf	- Developmental toxicityoxidative stress - Inflammation	[92]
	Sulfamethoxazole, sulfapyridine, sulfadiazine, sulfameter and sulfamerazine	0, 20, 40, 80 and 160 μg/L	24 and 168 hpf	Embryos	- Developmental toxicity - Cardiotoxicity	[93]
	Sulfamethazine	0, 0.2, 20 and 2000 μg/L	120 hpf	Embryo	- Developmental toxicity - Cardiotoxicity - Oxidative stress - Lipid peroxidation	[94]
			1: 7 days exposure, 7 days post-exposure 2: 1 day exposure, 2 days resting and 4 days re-exposure	Adult		
Tetracyclines	Oxytetracycline	0, 0.1, 10 and 10,000 μg/L	2 months	Adult	- Behavioral changes (boldness and hyperactivity) - Impairments at biochemical level - Alteration of bacterial communities of fish gut	[25]
		80 and 100 mg/kg body weight per day	6 weeks	Adult	- Higher digestive enzyme activities - More oxygen consumption rate	[41]
		1 and 5 ng/L	From 2 hpf to 120 dpf	Embryo	- Thyroid dysfunction - Developmental toxicity	[95]
		0, 75, 100, 150, 300, 600 and 900 mg/L	96 h	Embryos	- Developmental toxicity - Oxidative stress	[23]
		0, 1, 10, 25, 50 and 100 mg/L		Adult		
		0, 0.05, 0.5 and 5 mg/L	48 h	72 hpf	- Oxidative stress	[96]
		420 ng/L	6 weeks	Adult	- Impair the gut health - Higher metabolic rate	[91]
		10 μg/L	5 days and 2 months exposure 5 days and 1 month of post-exposure	Adult	- The effects are partially reversible	[43]
	Tetracycline HCl	0, 1 and 100 μg/L	1 month	Juvenile	- Body weight increase - No change in the body length	[42]
		0, 0.1, 1, 10, 100 and 1000 μM	48 h until 5 dpf	3 dpf	- Hepatotoxicity	[81]
		0, 2, 10, 20, 200, 2000 and 20,000 μg/L	96 h	4 hpf	- Developmental toxicity - Oxidative stress	[44]
	Minocycline	10–1000 ng/mL	3 days at 6–72 hpf and at 3–6 dpf	Embryos at 6 hpf and larvae at 3 dpf	- Fish died at 0.4 mg/mL	[54]
	Chlortetracycline	0, 6.25, 12.5, 25 mg/L	96 h	Adult	- Behavioral changes	[34]
		0, 0.2, 2 and 20 mg/L	48 h	72 hpf	- Oxidative stress	[96]
		0, 6.25, 12.5, 25 mg/L	96 h	Adult	- Behavioral changes	[34]
Polyether antibiotics	Maduramicin	0, 0.1, 0.5 and 2.5 mg/L	14 days	Adult	- Oxidative stress - Tissue damage in the gill, liver and intestine	[22]
		0, 10, 11.89, 14.14, 16.81 and 20 mg/L	96 h	Adult	- Impact on the survival - LC_50_ = 13.568 mg/L	
	Monensin	4, 4.34, 4.7, 5.1,5.53 mg/L	96 h	Adult	- LC_50_ = 4.76 mg/L	[39]
Others	Ceftazidime	10–1000 ng/mL	3 days at 6–72 hpf and at 3–6 dpf	Embryos at 6 hpf and larvae at 3 dpf	- Cardiotoxicity	[54]

Abbreviations: dpf = days post-fertilization, hpf = hours post-fertilization.

### 2.1. Aminoglycosides

Aminoglycosides are an important class of antibiotics that are widely prescribed in the treatment of Gram-negative bacterial infections (e.g., *Escherichia coli*, *Pseudomonas aeruginosa* and *Klebsiella pneumonia*) but at the same time have the potential to generate serious toxicities in patients. Unfortunately, in recent years, the use of aminoglycosides has increased because of the rising resistance of the other classes of antibiotics [97,98]. Studies on the toxicological effects of aminoglycosides in zebrafish showed different types of toxicity, including cardiotoxicity, ototoxicity, oculotoxicity and nephrotoxicity [97,99]. For instance, netilmicin, which is usually used in the treatment of various types of infections (e.g., urinary infections, lower respiratory tract infections, skin infections, septicemia, etc.) [100], induced cardiotoxicity in 6 hpf zebrafish embryos, but not for 3 dpf zebrafish, suggesting a toxic effect during embryonic development. The teratogenicity observed was mild and expressed as distorted heart structures and abnormal abdomens. Moreover, an increase in deaths corresponding with the increase in antibiotic dose was also observed (7, 10, 15, 20, 30 mg/mL) [54]. Gentamicin is another aminoglycoside with great bactericidal activity used for the treatment of over 40 clinical conditions but also for the manufacturing of various medical materials. The clinical experience of gentamicin use showed that it is an important player in the treatment of infectious and non-infectious diseases [101]. Studies conducted on zebrafish regarding the toxicity of this antibiotic showed that its administration can produce nephrotoxicity, oculotoxicity and ototoxicity and can even affect their locomotor behavior. For instance, in a study where larval zebrafish were injected with 10 mg/mL gentamicin, several features of acute renal failure were observed: reduction in the glomerular filtration rate, tubular obstruction, debris accumulation in the tubular lumen and lysosomal phospholipidosis [53]. Moreover, gentamicin treatment induced the loss of neuromast hair cells in 5-day-old zebrafish, demonstrating its otoxicity [75]. When more than 70% of the hair cells were lost, Han et al. [33] also observed a change in their locomotor behavior, manifested by latency of the startle response or by delayed reaction times. The oculotoxic effect of gentamicin was demonstrated in zebrafish larvae treated with 10 μM, from 3 to 5 dpf, by reduced optokinetic and visual motor responses [76]. An aminoglycoside, which can induce ototoxicity and also muscle damage in zebrafish, is neomycin. Zebrafish larvae of 5 dpf exposed for one hour at 125 μM neomycin showed a significant reduction in the number of hair cells and also a change in their swimming behavior related to a decrease in the total traveled distance or in the velocity and activity ratios [33]. Streptomycin, the first member of the aminoglycoside family, is known to have the lowest toxicological effect among this group of drugs [102]. However, studies on zebrafish showed ototoxicity and dysbiosis. A concentration of 400 μM streptomycin caused an intermediate ototoxicity, with a hair cell death rate of about 40–50% [77]. Moreover, the exposure to low concentrations of streptomycin (0.1–10 μg/mL) led to changes in the microbial diversity of zebrafish larvae, which consisted in the transition from the healthy mixture of Proteobacteria and Bacteroidetes to the dominance of Proteobacteria and, thus, early death [78]. Etimicin, a new aminoglycoside antibiotic, with good antimicrobial activity against Gram-positive and Gram-negative bacteria, shows low nephrotoxicity and ototoxicity in zebrafish embryos compared with gentamicin and amikacin [79]. 

### 2.2. β-Lactams

β-Lactam antibiotics are among the most prescribed drugs to treat medical conditions in humans due to various reasons, including their bactericidal effect against Gram-positive bacteria, at concentrations that overlap with those used for growth inhibition, the broad spectrum of activity and their lack of toxicity [102,103,104,105]. However, recent studies conducted on zebrafish showed that antibiotics of the β-lactam class can induce various types of toxicity. For instance, the exposure of young zebrafish to a high dose of amoxicillin (100 mg/L, for 7 days) caused locomotor alteration, a decrease in social interaction compared to the control group and an increase in oxidative stress [24]. Apart from the decrease in sulfhydryl content, indicating protein damage, there was a significant decrease in catalase activity and an increase in superoxide dismutase activity. Amoxicillin was observed to induce toxicity in zebrafish not only in the case of long-term exposure but also in the case of short-term exposure. Therefore, when the embryos and adults of zebrafish were exposed for 96 h to amoxicillin (at concentrations between 0–1125 mg/L and 0–100 mg/L, respectively), developmental and enzymatic biomarker abnormalities were observed. Premature hatching, inhibition of the catalase activity and induced glutathione-S-transferases in zebrafish adults were some of the issues reported [23]. Another β-lactam antibiotic that was reported to have toxic effects in zebrafish is cefotaxime. The exposure to this antibiotic for 1 day affected the locomotor behavior of the fishes and changed the expression of different genes related to the nervous system and sensory organs. For instance, the swimming time was observed to decline; furthermore, the regulation of neurotransmitter levels, synapse, neuron parts and sensory perception was affected [80]. A longer exposure of 120 h to 100 μg/L cefotaxime determined a decrease in the body lengths of zebra fish larvae, an increase in the activity of reactive oxygen species and different changes in the mRNA of various genes [83]. Several toxic effects were also observed in the exposure to ceftazidime. Cardiotoxicity, poor development of the bladder, delayed reactions, cognitive decline and increased aggressivity were a few of the effects recorded [34,54].

### 2.3. Macrolides

Macrolides are a class of antibiotics with inhibitory but not bactericidal effects against Gram-positive bacteria and also against some spirochetes or mycobacteria [102,104]. Despite the resistance developed by bacteria to this class, the antibiotics in this group are the second most prescribed drugs for the treatment of infectious and non-infectious diseases. The most commonly used antibiotics are erythromycin, clarithromycin and azithromycin [102]. Studies conducted on zebrafish larvae regarding the toxicity of these antibiotics and also of midecamycin, josamycin and acetylspiramycin showed hepatotoxicity manifested by liver degeneration, liver necrosis, hepatomegaly or steatosis [81,82]. Erythromycin, in addition to hepatotoxicity, depending on the dose and the zebrafish stage of development, can affect swimming ability by reducing the gene expression related to energy metabolism in the muscle when it is used in high concentrations (2 mg/L), can enhance it when its concentration is low (1–10 μg/L), may also produce endocrine disruption, delay the hatching, decrease the survival rate, increase or decrease the heart rate or up-regulate the genes related to catalase and superoxide dismutase activity [27,29,68]. Yan et al., in a study done on zebrafish embryos exposed to azithromycin, clarithromycin, tilmicosin and tylosin, reported cardiotoxicity manifested by tachycardia or bradycardia, depending on the drug concentration. Tilmicosin also caused cardiac congestion, pericardial edema and spinal curvature; decreased the activity of superoxide dismutase; increased the content of malondialdehyde; and up-regulated the pathways of apoptosis [36].

### 2.4. (Fluoro) Quinolones 

The (fluoro) quinolones, known for their four generations of antibiotics, have broad-spectrum activity against both Gram-negative and Gram-positive bacteria. Their mechanism of action mainly refers to the inhibition of DNA replication. Due to their high efficiency, this class of antibiotics is widely prescribed for the treatment of many human and veterinary bacterial infections [106]. Toxicity studies showed that (fluoro) quinolones can exhibit various effects on zebrafish embryos, larvae and also adults. Thus, Han et al. [86] evaluated the toxicity of 12 different (fluoro) quinolones (levofloxacin, gatifloxacin, norfloxacin, sparfloxacin, gemifloxacin, enoxacin, pefloxacin, prulifloxacin, lomefloxacin, moxifloxacin, ciprofloxacin, antofloxacin, 2-methylpiperazine, n-methylpiperazine and 4-n-Boc-2-methyl-piperazine) on zebrafish embryos and reported for all the antibiotics and concentrations (0.1–20 mM) tested, except 5 mM levofloxacin, head deformation, shortened tail, tail tip malformation, scoliosis and spinal curvature, reduced pigmentation, developmental retardation, pericardial edema, heart malformation and death. More than 80% of the deaths were related to cardiac malfunction, which manifested differently (heart rate and looping, cardiac hypogenesis and abnormal structure with evident edema) depending on the antibiotic. In another study, Xi et al. found that the exposure of embryos to norfloxacin induced neurotoxicity manifested by the inhibition of glial cell marker expression and increased stem cell marker and mature neuron marker expression. In addition, they observed an impaired hatching rate and enhanced mortality and malformation [85]. The effects of (fluoro) quinolones on zebrafish larvae consisted of increased expression of different important immune system cytokine genes caused by the exposure to 5 μg/L of ciprofloxacin or shortened body lengths, deformed body shape and disrupted metabolic processes as a result of 100 μg/L enrofloxacin action [83,87]. Regarding the toxicity of this class of antibiotics on zebrafish adults, Zhao et al. [88] showed that ofloxacin and enrofloxacin (5 μg/L) bioaccumulated in the liver, skin, muscles and gills but were not capable of triggering metabolic stress in the exposed fish. Moreover, in this study, it was observed that metallic ions, such as copper, could potentiate the accumulation of both drugs when co-administered in concentrations of 2.56 and 25.6 μg/L for 30 days. Cardiac toxicity in mature zebrafish was also reported by Shen et al. [89] in the case of ciprofloxacin- and gatifloxacin-exposed zebrafish. The authors observed certain dose-dependent cardiac dysfunctions, such as a decreased heart rate, which implies changes in the calcium-dependent signaling.

### 2.5. Sulfonamides

Compared with (fluor)quinolones, sulphonamides mainly suppress microorganism growth and multiplication and act as a competitive inhibitor of the bacterial enzyme dihydropteroate synthetase (DPS), an enzyme involved in folate biosynthesis [107]. Tokanová et al. [90] have showed that the exposure of zebrafish to sulfamethoxazole (50–500 μg/L) for 14 days has no significant impact on the antioxidant enzyme activity (e.g., glutathione peroxidase, glutathione reductase and glutathione S-transferase) compared to the control group. In the same study, a slight increase in lipid peroxidation has been observed for all tested concentrations. On the other hand, long-term exposure to sulfamethoxazole (260 ng/L) did not influence the weight gain of zebrafish but increased the metabolic rate [91]. In contrast with previous studies, Yan et al. [84] have reported that the body weight and the egg production were depressed by sulfamethoxazole at 200 μg/L in a partial life cycle study with zebrafish. However, the activities of antioxidant (superoxide dismutase and catalase) and metabolic (ethoxyresorufin O-deethylase) enzymes were stimulated. The results revealed that parental exposure to sulfamethoxazole at 200 μg/L could impact the optimal development of the next generation, leading to a decreased hatching and survival rate and enhanced developmental abnormalities. An increase in fish body growth was also observed after exposure to sulfamethoxazole and sulfadiazine (160 μg/mL) and to sulfamethoxazole–sulfadiazine and sulfamerazine–sulfameter binary mixtures, whereas a decrease in fish body weight was observed for sulfapyridine (20 μg/L) and sulfamerazine (40 μg/L). Similar observations were made for heart rate changes; thus, the sulfapyridine and sulfameter (80 μg/L) increased the cardiac activity, whereas sulfadiazine, sulfamethoxazole and sulfamerazine led to no visible effects [93]. Additionally, inflammation and immune responses have been observed among young healthy zebrafish [92]. The up-regulation of mRNA levels of several key proinflammatory cytokines and their corresponding proteins, including interleukin-1β (IL-1β), interferon-γ (IFN-γ) and interleukin-11 (IL-11), was observed after 120 h of sulfamethoxazole exposure, as well as the interleukin-6 (IL-6) and tumor necrosis factor-α (TNF-α) displaying a dose-response relationship. Interestingly, Chen et al. [94] considered two lifespan stages (embryo and adult) and three exposure periods (exposure, post- and re-exposure) in order to analyze the effects induced by sulfamethazine. The results suggested that sulfamethazine can induce side effects in both embryos and adult zebrafish. For example, the antibiotic exposure was able to cause delays in the hatching of embryos at 56–96 hpf, edema, increased heartbeats and body impairments (e.g., reduced the body length and spinal curvature). Moreover, both the superoxide dismutase activity and malondialdehyde content of the adult zebrafish in the exposure and re-exposure periods changed. Additionally, the authors mentioned that lipid peroxidation was reversible following adult zebrafish sulfamethazine exposure, while re-exposure has an additive effect on the oxidative stress status. However, Chen et al. [108] have proved that the presence of sediment particles in the water and the high salinity can reduce sulfamethoxazole accumulation in zebrafish tissues.

### 2.6. Tetracyclines

Tetracyclines are a broad-spectrum antibiotic class, effective against both Gram-positive and Gram-negative bacteria. Compared with the other antibiotic groups, they differ by mechanism of action through the inhibition of protein synthesis. Thus, the drug molecules link to the 30S ribosomal subunit of the bacteria and prevent the connection of specific components necessary for bacterial synthesis and growth [25,34]. Almeida et al. [25], by evaluating the long-term effects of oxytetracycline exposure in zebrafish, have found that the metabolic rate of the exposed fish and their behavior were significantly altered. The results revealed that low oxytetracycline concentrations were followed by increased exploratory behavior, while the bacterial communities of the fish gut were affected by the highest concentration of the antibiotic. Moreover, the swimming pattern changes observed in the zebrafish were associated with photosensibility induced by antibiotic treatment. Effects were observed at the biochemical level, including oxidative damage (reduced levels of total glutathione, glutathione S-transferase and catalase) and increased energy consumption. On the other hand, based on the higher similarity between exposed and control groups, they demonstrated that the effects induced oxytetracycline exposure at the physiological and microbiome levels are partially reversible after a recovery period (5 days and 1 month) [43]. Zhang et al. [44], by studying the effects of tetracycline on developmental toxicity and molecular response in zebrafish embryos, have demonstrated that the antibiotic exposure can induce oxidative stress and cell apoptosis, which determine a developmental delay. Thereby, tetracycline was found to interfere with zebrafish growth and development, displaying delayed hatching, decreased body length, increased yolk sac area and absence of a swim bladder. The gene expression pattern demonstrates that caspase-dependent apoptotic pathways may contribute to tetracycline-induced apoptosis in the early life stages of the zebrafish. Moreover, Keerthisinghe et al. [42] showed that chronic long-term exposure to tetracycline could determine body weight increase in adult zebrafish through hepatic lipid metabolism impairment. In addition, the analysis revealed that tetracycline could alter the microbial community composition of the zebrafish gut and increase the bacteria diversity. Similarly, Zhou et al. [41] have proved that exposure to oxytetracycline can induce adverse effects in zebrafish gut health. Meanwhile, Yu et al. [95] revealed that long-term exposure to oxytetracycline can cause thyroid dysfunction and affect the growth and development of zebrafish. Both the triiodothyronine (T3) and thyroid-stimulating hormone (TSH) contents were altered; therefore, after treatment, the T3 contents were significantly enhanced and TSH secretion was reduced. A recent study showed that a high dose of tetracycline (100 μg/L) has the potential to inhibit the growth of zebrafish larvae [83]. Furthermore, acute exposure to a high dose of oxytetracycline (72h-EC50 = 127.6 mg/L) can cause delayed hatching of zebrafish embryos, and treatments with oxytetracycline (0.05 mg/L) and chlortetracycline (20 mg/L) reduce the glucose levels in zebrafish larvae [96]. Exposure to oxytetracycline (0.42 mg/L) for a six-week period increased the metabolic rate and disturbed the intestinal microbiota [91]. Changes in the locomotor behavior, memory/learning processes and aggressive behavior were observed after acute exposure of the fish to chlortetracycline. The results indicate that locomotor impairment appears shortly after drug exposure [34]. 

### 2.7. Polyether Antibiotics

The polyether antibiotic class usage emerged from the need to renew the broad-spectrum antibiotic pool due to the development of many multiresistant bacterial strains. Active against both Gram-positive and Gram-negative bacteria, the carboxylic ionophores’ mechanism of action consists of their ability to increase the ion permeability of the cellular membranes [109]. Considering this aspect, the possible toxic effects could be exhibited mainly in the mitochondria leading to energetic cellular impairment. Further effects of this mechanism could reflect in the muscle tissues, including myocardial tissues, by degeneration [110]. Ni et al. showed that after 3 days of exposure to 0.1 mg/L and 2.5 mg/L of maduramicin, the activities of antioxidant enzymes (e.g., superoxide dismutase, catalase, glutathione peroxidise and glutathione s-transferase) in the liver of adult zebrafish significantly changed; they were increased for the lower concentration and were inhibited for the higher concentration. Moreover, the authors observed that the subacute exposure to 2.5 mg/L of maduramicin can induce severe oxidative stress and tissue damage in the gill, liver and intestine of zebrafish, without impact on their survival [22]. Li et al. [39], studying the chronic toxicity of monensin against zebrafish, reported a medium toxicity with an LC50 (96 h) value of 4.76 mg/L, which is comparable with oxytetracycline and norfloxacin.

## 3. Effects of Antibiotic Mixtures in Zebrafish

Considerable efforts have been devoted to the toxicity of single antibiotics, although in aquatic environments, the organisms are not just exposed to a single antibiotic but rather to a mixture of various antibiotics, which may interact with each other. Consequently, the evaluation of the toxic effects of single antibiotics does not offer a close-to-reality view of their impact against aqueous ecosystems. For example, their cumulative effects may be antagonistic, additive or synergistic [111]. Zhang et al. [37], studying the joint toxicity of fluoroquinolones (FQs) and tetracyclines (TCs) on the zebrafish embryo-larval, found that the toxicological effects of combined TCs–FQs were comparable or slightly less than those of TCs alone. They demonstrated that TCs played a key role in the mixtures, and their simultaneous presence mainly had antagonistic actions. A similar result has been reported by Ding et al. [38], with the TCs having a stronger effect than the DKAs, as the 120 hpf LC50 value for the malformation rate due to the exposure to the antibiotic mixtures was 149.8 mg/L, and for TCs, it was 18.3 mg/L. In contrast with previous results, the joint antibiotic toxicity of sulfamonomethoxine, cefotaxime sodium, tetracycline and enrofloxacin was more severe than that of a single treatment [83]. Moreover, the binary mixtures of sulfonamides showed elevated developmental toxicity and a higher impact to the detoxification pathway in zebrafish embryos [93]. 

As can be seen in Table 4, few data are available on the effects of antibiotic mixtures using zebrafish as a model organism. Wang and co-workers provided important information concerning the joint effects of β-diketone antibiotic mixture exposure on zebrafish. They demonstrated that the exposure to fluoroquinolones and tetracyclines can induce dysfunction of the immune system, liver and heart; damage the antioxidant defense system; and cause reproductive failure, skeletal disorders and neurotoxicity effects [30,31,35,37,112,113,114]. Briefly, treatments with DKAs affect a variety of cellular and biological processes in zebrafish. It is very interesting to note that only few data are available on the neurotoxicity of antibiotics from the perspective of animal behavior. In a recent study, the effects of DKA chronic exposure on zebrafish behavior were examined by means of a bottom swelling test, conditioned place preference (CPP) and social cohesion using Ethovision XT software Noldus IT, as well as their related molecular mechanisms [28]. The video-tracking analysis revealed that DKA exposure significantly impaired zebrafish swimming, exploratory behavior and social interaction. According to the bottom swelling test, the number of transitions to the upper portion, the time spent in the upper side and the distance traveled by zebrafish exposed to 6.25, 12.5 and 25 mg/L DKA treatments were significantly affected. While zebrafish exposed to 6.25 mg/L of DKAs showed an increased exploratory ability, indicating occurrence of anxiolytic behavior, the 25 mg/L DKA treatment decreased the time spent in the top part of the test tank when compared to the control group, which may be interpreted as an indicator of anxiety-like behavior (Figure 1). The distance traveled was enhanced by 85% and the mean speed by 87% for 6.25 mg/L DKA treatment compared to the control. The zebrafish active levels were prominently decreased with the increase in DKA concentrations from 6.25 mg/L to 25 mg/L. Moreover, an increase of 38% in zebrafish shoaling behavior was observed, which may be due to anxiety-like shoal cohesion or increased sociability, and there was a decrease of 41% in zebrafish social cohesion, possibly due to an autism-like state, for the low DKA concentration. In agreement with behavioral abnormality, the TEM images confirmed that DKA exposure may inhibit the development of primary motor neurons, such as pyknosis of the nucleolus, cavitation of cytoplasm, decreasing the synapse distance and dissolution of filament in axon. In addition, the treatment with the β-diketone antibiotic mixtures led to changes in the transcriptional levels of 11 locomotor-related genes. Similar to those reported previously, Zhang et al. [37] showed that the circadian cycle was disturbed after the exposure of zebrafish to DKAs.

Based on high-throughput screening and histopathological observations, joint DKA exposure on larval and adult zebrafish resulted in abnormal expression of a large number of micro-RNAs; some long non-coding RNAs and their regulating target genes; and a series of physiological changes in tissues and organs, especially those related to the nervous, immune and reproductive systems [32,112,113,114,115,116]. For example, the joint FQ and TC exposure induced severe histopathological changes and damage in the zebrafish eye, brain, hepatic and spleen tissues (Figure 2), including vacuolation of interstitial cells, photoreceptor cell cysts, reduced number of neurons, glial cell proliferation, formation of glial scars, hepatic parenchyma vacuolar degeneration and brown metachromatic granules. On the other hand, the female reproductive system suffered serious damage after chronic exposure to DKAs, as reflected by the increased number of early developmental oocytes, irregular cell distribution, different cellular morphology, decreased yolk granules, cytoplasmic shrinkage and cell lysis in mature oocytes [32]. Moreover, Wang et al. [114] showed the adverse reproductive effects of β-diketone antibiotics in female zebrafish, along with an effect transfer relation across parents and their offspring. A negative developmental impact of offspring was observed. This observation is consistent with the recently published results of Qiu et al. [116], suggesting that gravid female zebrafish exposure to environmentally relevant concentrations of antibiotic mixtures may disturb maternal and offspring health (Table 2). After 4 weeks of treatment, an accumulation of low concentrations of antibiotics was detected in the F0 ovary and F1 eggs, indicating an antibiotic transfer from exposed adult fish to their offspring. The chlortetracycline (358.4 ± 20.7 ng/g), enrofloxacin (298.5 ± 34 ng/g) and tetracycline (256.1 ± 19.1 ng/g) were the predominant antibiotics accumulated in the F1 eggs following the maternal exposure to 100 μg/L of the antibiotic mixtures for 4 weeks. Moreover, a relationship between the accumulation of antibiotics and the morphological effects in fish embryos and larvae and the maternal exposure time and antibiotics concentrations was demonstrated, suggesting that long-term exposure and higher concentrations of chemicals might lead to a greater environmental health risk. Furthermore, the exposure of female zebrafish to a mixture of 15 commonly detected antibiotics in the environment during the gravid period might induce impairment of gastrointestinal function in F1 offspring.

Additional studies proved that treatments with β-diketone antibiotic mixtures led to abnormal development of otoliths in zebrafish embryos and a decreased sensitivity to acoustic stimulation, which implies the occurrence of hearing impairment [31]. Notably, the miR-96 and miR-184 were identified as key factors regarding the molecular mechanisms regulating hearing functions. Whereas the miR-184 is involved in construction of the otic vesicle during zebrafish embryonic growth, the miR-96 plays an important role in otic vesicle development and the formation of hearing. It has to be highlighted that the previous observations help to provide a better understanding of the mechanisms behind hearing loss due to antibiotic exposure, as well as to provide theoretical guidance about early intervention and gene therapy for drug-induced diseases. Based on the experimental and omics analyses, Qiu et al. [83] proved that the metabolic pathways of zebrafish larvae were affected by treatment with the antibiotics sulfamonomethoxine, cefotaxime sodium, tetracycline and enrofloxacin at concentrations of micrograms per liter. Moreover, the short-term exposure to the four antibiotics affected the development of zebrafish larvae. Yin et al. [30], by using the two-dimensional gel electrophoresis and MALDI-TOF-MS techniques to analyze zebrafish protein expression after long-term DKA exposure, found that 47 protein spots have a greater than twofold differential expression compared to the control, and the number of positive proteins was 26, with 14 proteins up-regulated and 12 proteins down-regulated. The authors assumed that the main functions of the differentially expressed proteins were involved with skeletal and cardiac muscles, signal transduction and energy transfer. On the other hand, they found that DKA exposure can cause a significant decrease in the heart rate of embryonic zebrafish, without abnormalities in cardiac shape, the reduction being proportional with an increase in antibiotic concentrations [30,37,117]. 

As a result of long- or short-term exposure to a mixture of antibiotics, the balance between the action of reactive oxygen species (ROS) and antioxidant defense systems may be impaired, leading to the generation of oxidative stress in zebrafish. Thus, alterations of enzymatic (e.g., SOD) and nonenzymatic (e.g., GSH) antioxidant activities have been reported as a result of the zebrafish self-protection against antibiotics toxicity [117]. Briefly, the antibiotic mixtures induced glutathione production at the end of the hatching period and significantly inhibited superoxide dismutase activities [35]. In accordance with the results of the previous study, Qiu et al. [83] indicated that treatment with sulfonamide β-lactamtetracycline and quinolone antibiotics could affect the ROS content in zebrafish larvae. Changes in creatine kinase activity and creatinine concentration, which are related to muscle contraction and energy transfer, were observed in zebrafish after exposure to a series of DKA concentrations [38]. 

Recently, Bielen et al. [118] evaluated the antibiotic contamination of the final effluents from two Croatian pharmaceutical companies during four seasons and studied the effects of the antibiotic residues on zebrafish embryos. The results revealed that the effluents of Company 1 are a source of macrolide pollution, while sulfonamide, trimethoprim, enrofloxacin and oxytetracycline antibiotics were detected in the water samples collected from the region of Company 2 (Table 3). All tested samples, including the winter and spring effluent samples from Company 1 and the summer and autumn samples from Company 2, induced side effects on zebrafish embryos. Briefly, the most highly contaminated effluents caused the most pronounced abnormalities in fish. For example, during exposure to the Company 1 winter sample, embryos often exhibited heart and yolk edema, scoliosis and lack of pigmentation formation. Whereas the only observed effects during exposure to the Company 2 winter and autumn effluents were pericardial, yolk sack edema and an increased heart beat at 48 and 72 hpf.

Interestingly, Maselli et al. [119] revealed that the administration of ampicillin (100 μg/mL), kanamycin (100 μg/mL) and amphotericin B (250 ng/mL) over two weeks to zebrafish with short bowel syndrome resulted in less intestinal inflammation and reduced liver steatosis after alteration of the intestinal microbiome (Table 4).

**Table 4 pharmaceuticals-14-00578-t004:** Overview of the potential biological effects of antibiotics represented as mixtures in zebrafish, dating from 2014 to the present.

Class	Antibiotics	Mixture Concentrations (Weight/Water Volume)	Exposure Period	Lifespan Stages	Effects Observed	Refs.
β-Diketones (fluoroquinolones and tetracyclines)	Ofloxacin, ciprofloxacin, enrofloxacin, doxycycline, chlortetracycline and oxytetracycline	0, 6.25 and 12.5 mg/L	From embryos (4 hpf) to larvae (90 dpf) stage	Larvae and adult	- Abnormal expression of differentially expressed miRNAs - Vacuolation of interstitial cells, reduced number of neurons, glial cell proliferation and deformation of glial scar	[113]
		0, 12.5 and 25 mg/L	From embryos (2hpf) to larvae (5dpf) stage	72 hpf or 120 hpf	- Ototoxicity	[31]
		0, 6.25 and 12.5 mg/L	3 months	90 dpf	- Immunotoxicity (abnormal expression of immune genes and enzymes and variable levels of damage to immune-related organs)	[112]
		0, 6.25, 12.5 and 25 mg/L	3 months	Embryos at 6 hpf	- Neurotoxicity (behavioral abnormality and anxiety, pathological changes of nerve cells, changes in *appb* and *cdh6* transcriptional level)	[28]
	FQs: ciprofloxacin, ofloxacin, norfloxacin, enrofloxacin,	0, 25, 50, 100, 200, 300, 400 and 600 mg/L	6–120 hpf	72 hpf or120 hpf	- Abnormal hatching - Mortality- Malformation	[37]
	TCs: chlortetracycline and doxycycline	0, 1.56, 3.13, 6.25, 12.5, 25 and 50 mg/L				
	FQs + TCs	0, 4.69, 9.38, 18.75, 37.5, 75, 150, 300 and 450 mg/L				
	FQs:	0, 12.5, 25, 50 mg/L	6–96 hpf	144 hpf	- Higher and basal swimming speed	
	TCs:	0, 1.56, 3.13, 6.25 mg/L				
	FQs + TCs	0, 4.69, 9.38, 18.75 mg/L				
	FQs + TCs	0, 9.38, 18.75, 37.5, 75, 150 mg/L	6–72 hpf	48, 60, 72 hpf	- Decreased heart rate	
	FQs + TCs	0, 45, 60, 95 mg/L	2–4 months	-	- Severe edema in sarcoplasmic reticulum, melted muscle fiber and edema in mitochondria (skeletal muscle) - Disordered arrangement of muscle fibers, melted fiber, partial edematous membrane cell nuclear materials, edema in mitochondria, abnormal mitochondria (heart) - Changes in transcriptional levels of *acta1a, myl7* and *gle1b* genes, which are involved in heart development and skeletal muscle formation	
	Ciprofloxacin, ofloxacin, enrofloxacin, oxytetracycline, chlortetracycline and doxycycline	0, 6.25, 12.5 and 25 mg/L	6 hpf until 144 hpf	90 dpfwild-type adult zebrafish	- Physiological impairment - Reproductive toxicity	[114]
	Ciprofloxacin, ofloxacin, norfloxacin, enrofloxacin, chlortetracycline and doxycycline	0, 18.75, 37.5, 75, 150, and 300 mg/L	From 6 hpf to 120 hpf	Studied every 12 h using a microscope	- Abnormal hatching - Malformation and mortality - Decreased heart rate	[117]
		0, 2.34, 9.38 and 37.5 mg/L	From 6 hpf to 96 hpf	Embryos (<72 hpf) and larvae (>72 hpf)	- Locomotor toxicity - Oxidative stress (SOD and GSH)	
	Ciprofloxacin, ofloxacin, enrofloxacin, oxytetracycline, chlortetracycline and doxycycline	0, 6.25, 12.5 mg/L	From 6 hpf to 90 dpf	90 dpf	- Physiological impairment - Reproductive toxicity	[32]
					- Abnormal expression of some lncRNAs and their regulating target genes - Liver and spleen toxicity	[115]
	Ciprofloxacin, ofloxacin, norfloxacin, enrofloxacin, chlortetracycline and doxycycline	0, 9.38 mg/L	From 6 hpf to 144 hpf	90 dpf	- 47 differential expression proteins vs. control with 14 up-regulated and 12 down-regulated	[30]
		0, 4.69, 9.38, 18.75 and 37.5 mg/L	From 6 hpf to 96 hpf	120 hpf	- No visible developmental malformation - Greater spontaneous movement for low dose	
		9.38, 45 and 60 and 90 mg/L	From 6 hpf to 90 dpf	90 dpf	- Changes in creatine kinase activity and creatinine concentration - Changes in heart tissue reflected by dissolution of cristae and vacuolation of mitochondria	
		0, 9.38, 18.75, 37.5, 75 and 150 mg/L	From 6 hpf to 72 hpf	48,60 and 72 hpf	- Decreased heart rate	
		0, 18.75, 37.5, 75, 150, 300 and 450 mg/L	From 6 hpf to 120 hpf	72 hpf120 hpf	- 72 hpf EC50 for hatching rate = 130.3 mg/L - 120 hpf EC50 for malformation rate = 135.1 mg/L - 120 hpf LC50 for malformation rate = 149.8 mg/L - Severe malformation	[38]
		0, 45, 60 and 90 mg/L	60 dpf	7, 14, 21 days	- Changes in creatine kinase activity and creatinine concentration	
		9.38 mg/L	From 6 hpf to 6 days	90 dpf	- Serious liver damage	[35]
		2.34, 9.38 and 37.5 mg/L			- Oxidative stress (SOD and GSH)	
Sulfonamides (binary mixtures)	Sulfamethoxazole, sulfapyridine, sulfadiazine, sulfameter and sulfamerazine	0, 20, 40, 80 and 160 μg/L equi-toxic ratio	24 and 168 hpf	Embryos	- Developmental toxicity - Cardiotoxicity	[93]
Sulfonamides, β-lactams, tetracyclines and quinolones	Sulfamonomethoxine cefotaxime sodium tetracycline enrofloxacin	0.01, 1 and 100 μg/Lin equal proportions	120 h	Embryos at 4 hpf	- No significant differences in the mortality - Decreased the body lengths - Changes in the mRNA transcription profiles	[83]
Macrolides, amphenicols and sulfonamides	Clarithromycin, florfenicol, sulfamethazine	0.1 mg/L	96 h	Embryo	- No visible morphological changes - Behavioral and metabolic effects	[40]
			-	5 dpf		
			72 h	Adult		
Macrolides, lincosamides, quinolones, sulfonamides, tetracyclines and other	Clarithromycin, erythromycin, roxithromycin, lincomycin, ciprofloxacin, enrofloxacin, norfloxacin, ofloxacin, sulfadiazine, sulfamethazine, sulfamethoxazole, trimethoprim, oxytetracycline, chlortetracycline and tetracycline	0.1 and100 μg/L	4 weeks	150 dpf gravid fish	- Reproductive effects - An antibiotic transfer from exposed adult fish to their offspring - Gastrointestinal effects in zebrafish offspring	[116]
β-lactams, aminoglycosides and macrolides	Ampicillin, kanamycin and amphotericin B	Mixture of 100 μg/mL AMP, 5 μg/mL KAN 250 ng/mL AMB	2 weeks	Adult male zebrafish	- Alter the intestinal microbiome - Decrease intestinal and hepatic inflammation - Decrease hepatic steatosis in zebrafish with SBS	[119]

Abbreviations: TEM = transmission electron microscopy, ROS = reactive oxygen species, hpf = hours post-fertilization, dpf = days post-fertilization, FQs = fluoroquinolones, TCs = tetracyclines, EC50 = half maximal effective concentration, LC50 = lethal 50, AMP = ampicillin, KAN = kanamycin, AMB = amphotericin B, SBS = short bowel syndrome.

## 4. Conclusions and Future Perspectives

As a result of the limitations of both traditional mammalian models and cell-based assay methods, in recent years, zebrafish have rapidly gained acceptance by the scientific community as a promising animal model to study the effects induced by various drugs. According to previous studies, zebrafish can be used to predict several drug-related side effects. Moreover, as a result of the acute and chronic exposure to antibiotics, the zebrafish health or/and its capacity to respond to environmental changes are affected. Single or combined antibiotic exposure may lead to zebrafish dysfunction and the occurrence of diseases related to the cardiovascular, nervous, digestive and immune systems. To date, few data are available on the neurotoxicity of antibiotics from the perspective of animal behavior. For proper drug testing, the combined effects of multiple antibiotics and the interaction of the antibiotics with other chemical compounds should be thoroughly studied. Moreover, to avoid potentially false-positive or negative results, it is necessary to quantify the amount of antibiotics absorbed by each zebrafish specimen. Therefore, the development of a simple, rapid, cheap and sensitive analytical technique for chemical quantification in fish tissues remains a challenge for researchers. Further research is required on the molecular mechanisms in order to establish a relationship between the molecular events and behavioral, physiological and morphological effects, as well on the capacity of zebrafish to recover at the physiological, biochemical and microbiome levels after chemical exposure (post-exposure period).

## Figures and Tables

**Figure 1 pharmaceuticals-14-00578-f001:**
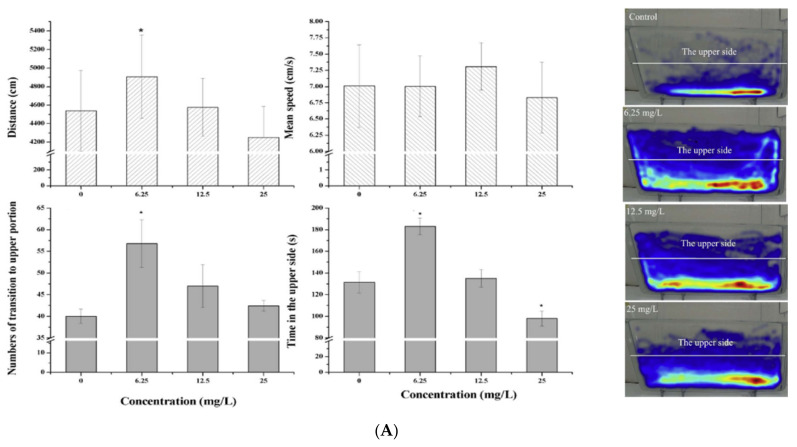
Basic behavioral endpoints of the control and DKA exposed groups (6.25, 12.5 and 25 mg/L) obtained from bottom swelling (**A**) and conditioned place preference (**B**) tests. The video trials resulting from the behavioral tests were reused in Ethovision XT software (Noldus IT, Wageningen, Netherlands) to generate average heat maps for experiments. Data represent average ± SD (*n* = 33–36 individuals for bottom swelling test and *n* = 34–36 individuals for CPP test) evaluated by one way ANOVA (* *p* < 0.05 and ** *p* < 0.01) [28].

**Figure 2 pharmaceuticals-14-00578-f002:**
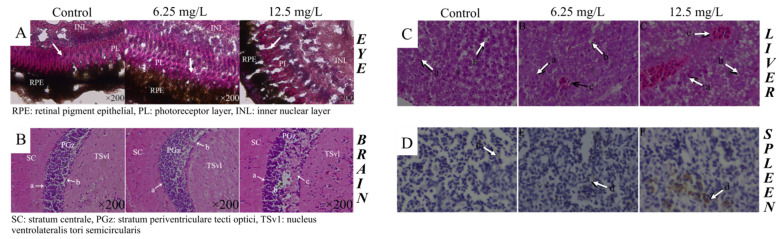
Histopathological changes in adult zebrafish eye (**A**), brain (**B**), hepatic (**C**) and spleen (**D**) tissues after exposure to a series of DKA concentrations. Arrows in Figure 2A indicate photoreceptor cell cysts and intercellular vacuoles; in Figure 2B the “a arrow” shows the decrease in neuron number, “b arrow” shows ventriculomegaly and “c arrow” shows glial cell proliferation and the formation of a glial scar; in Figure 2C the “a arrow” indicates reduced, swollen and vague hepatocytes, “b arrow” shows hepatic parenchyma vacuolar degeneration and “c arrow” denotes blood accumulation and clot formation; in Figure 2D the “d arrow” shows metachromatic granules. Histopathological observations on these organs were performed with hematoxylin and eosin (HE) staining using a standard protocol. The photographs of the tissues were taken when 3 biological replicates showed a high degree of uniformity. [113,115].

**Table 2 pharmaceuticals-14-00578-t002:** Morphological measurements for F0 female and F1 embryo and larval zebrafish following maternal antibiotic mixture treatment for 4 weeks. All data are expressed as mean ± SD. [116].

Changes in F0 Female (n = 20 Individuals)	Antibiotic Concentrations		
	Control	1 μg/L	100 μg/L
Body weight (g)	0.51 ± 0.016	0.53 ± 0.015	0.56 ± 0.025
Body length (cm)	3.58 ± 0.029	3.55 ± 0.032	3.66 ± 0.031*
Intestinal weight (g)	0.02 ± 0.001	0.025 ± 0.002	0.024 ± 0.001
Ovary weight (g)	0.07 ±0.004	0.084 ± 0.005	0.087 ± 0.004
**Changes in F1 embryo and larval**	**Control**	**1 μg/L**	**100 μg/L**
Egg production (number per parent, 20 individuals)	490.6 ± 23.09	442 ± 134.51	397.3 ± 31.39
Egg death rate at birth (% 0 hpf, 3 biological replicates)	5.09 ± 2.05	6.11 ± 3.05	17.76 ± 3.3 *
Fertilization rate (% 4 hpf, 3 biological replicates)	73.2 ± 0.73	78.6 ± 1.77	74.7 ± 2.25
Egg death rate at 120 hpf (%, 3 biological replicates)	1.7 ± 0.13	2 ± 0.26	5.9 ± 0.94 *
Hatching rate (%, 72 hpf, 3 biological replicates)	94.5 ± 2.94	92.1 ± 6	89.4 ± 3.29
F1 body length (mm, 120 hpf, 20 individuals)	3.91 ± 0.02	3.88 ± 0.02	3.92 ± 0.02
Displacement distance (mm, 0–10 min, 120 hpf, 20 individuals)	582.7 ± 106.2	678.4 ± 109	465.4 ± 58.6

Statistical significance defined as *p* < 0.05, indicated by an asterisk.

**Table 3 pharmaceuticals-14-00578-t003:** Macrolides concentrations measured in effluents of Company 1 and the concentrations of Figure 2. during four seasons [118].

	Antibiotic	Concentration (μg/L)	
		Winter	Spring
1	Azithromycin	2137	3776
	n-desmethyl azithromycin	2341	5660
	Erythromycin	2009	1069
		Summer	Autumn
2	Sulfadiazine	7.1	3
	Sulfamethazine	231	6.7
	Trimethoprim	5.4	1
	Enrofloxacin	4.3	3.6
	Oxytetracycline	29	10

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
