# Peer review of "Zebrafish as a Screening Model to Study the Single and Joint Effects of Antibiotics †"

_pharmaceuticals, 2021, doi:10.3390/ph14060578_

Round 1

Reviewer 1 Report

Review Report attached

Author Response

Reviewer 1

 Rating the manuscript:

In this manuscript the authors have implicated the hazardous impact of antibiotic wastes from pharmaceutical companies and human or animal excreta in sewerage systems. In this review the in vivo studies represent an essential step to study the potential impact induced by pharmaceuticals exposure. In this review the authors have summarized the recent advances made on the toxicity of seven representative classes of antibiotics, including aminoglycosides, -lactams, macrolides, quinolones, sulfonamides, tetracyclines and polyether antibiotics towards zebrafish, as well as the combined effects of antibiotic mixtures, to date. In this review, the authors have stressed on the role of antibiotic mixtures effects using zebrafish as an animal model in toxicity. They have also concluded that most of the research papers have focused on antibiotics toxicity towards zebrafish, across different developmental stages, rather than on their efficacy assessment. This work might not directly fall within the radius of the aims and scope of the journal, like small molecules as drug candidates, therapeutic tools, biological targets and biomarkers, radiopharmaceuticals sciences, radiochemistry and nuclear medicine, pharmacokinetics and pharmacodynamics, pharmaceutical preparations and drug delivery, however it raises a concern regarding the pharmaceutical drug residue accumulation in the environment and their potential harmful effects on ecosystems and human health and is of relevance to public health. This work marks as a caution to all pharmaceutical companies in delivering their wastes safely in the ecosystem and ensuring biodegradable pre-discharge procedures in manufacturing plants to reduce accumulation of antibiotic wastes. It also raises a concern over excess usage of antibiotics in medical formulations as therapeutic interventions and hints towards possible side effects in various biological processes if used in high doses and certain combinations.

Overall Recommendation:

Accept after Revisions: The paper is in principle accepted after minor revisions based on the reviewer’s comments. Review Report: However, there are few points that need to be addressed in the paper:

  1. Figure 1: needs bar graphs, or quantitative representation of heat maps and details of how many animals are present in this assay.

Our response: As suggested by the reviewer, we have revised the Figure 1 for clarity.

Changes in the manuscript:

(Page 16)

(A)

(B)

Figure 1. Basic behavioral endpoints of the control and DKAs exposed groups (6.25, 12.5 and 25 mg/L) obtained from bottom swelling (A) and conditioned place preference (B) tests. The video trials resulted from the behavioral tests were reused in Ethovision XT software (Noldus IT, Wageningen, Netherlands) to generate average heat maps for experiments. Data represent average ± SD (n=33-36 for bottom swelling test and n=34-36 for CPP test) evaluated by one way ANOVA ( p < 0.05 or p < 0.01). [28]

  1. Figure 2: Legends need to be detailed. I would like to see the details of the histopathological hallmarks, their names, full forms of the abbreviations, details of the staining etc. The figures should be detailed enough that the readers do not have to go back to the text. Scale bars needs to be added. Number of animals and details of section width should also be mentioned.

Our response: As suggested by the reviewer, we have revised the Figure 2 for clarity.

Changes in the manuscript:

(Page 17)

Figure 2. Histopathological changes in adult zebrafish eye (A), brain (B), hepatic (C) and spleen (D) tissues after exposure to a series of DKA concentrations. Arrows in Figure 2A indicate photoreceptor cell cysts and intercellular vacuoles; in Figure 2B the “a arrow” shows the decreased in neurons number, “b arrow” shows ventriculomegaly and “c arrow” shows glial cell proliferation and the formation of glial scar; in Figure 2C the “a arrow” indicates reduced, swollen and vague hepatocytes, “b arrow” shows hepatic parenchyma vacuolar degeneration and “c arrow” denoted blood accumulated and clot formatted; in Figure 2D the “d arrow” shows metachromatic granules. Histopathological observations on these organs were performed with haematoxylin and eosin (HE) staining using a standard protocol. The photographs of tissue were taken when 3 biological replicates showed a high degree of uniformity. [116, 118]

  1. Figure 3: Total number of animals tested, or total number of ovaries across n number of animals should be mentioned. Quantitative representation with standard error mean bars, with p value should be added in these figures.

Our response: As suggested by reviewer, we have revised the information for clarity.

(Page 18)

The chlortetracycline (358.4 ± 20.7 ng/g), enrofloxacin (298.5 ± 34 ng/g) and tetracycline (256.1 ± 19.1 ng/g) were the predominant antibiotics accumulated in the F1 eggs following the maternal exposure to 100 mg/L of the antibiotic mixtures for 4 weeks.

Table 2. Morphological measurements for F0 female and F1 embryo and larval zebrafish following maternal antibiotic mixture treatment for 4 weeks. All data are expressed as mean ± SD. [116]

Changes in F0 female

(n = 20 individuals)

Antibiotic concentrations

Control

1 mg/L

100 mg/L

Body weight (g)

0.51 ± 0.016

0.53 ±0.015

0.56 ± 0.025

Body length (cm)

3.58 ± 0.029

3.55 ± 0.032

3.66 ± 0.031*

Intestinal weight (g)

0.02 ± 0.001

0.025 ± 0.002

0.024 ± 0.001

Ovary weight (g)

0.07 ±0.004

0.084 ± 0.005

0.087 ± 0.004

Changes in F1 embryo and larval

Control

1 mg/L

100 mg/L

Egg production (number per parent, 20 individuals)

490.6 ± 23.09

442 ± 134.51

397.3 ± 31.39

Egg death rate at birth (% 0 hpf, 3 biological replicates)

5.09 ± 2.05

6.11 ± 3.05

17.76 ± 3.3*

Fertilization rate (% 4 hpf, 3 biological replicates)

73.2 ± 0.73

78.6 ± 1.77

74.7 ± 2.25

Egg death rate at 120 hpf (%, 3 biological replicates)

1.7 ± 0.13

2 ± 0.26

5.9 ± 0.94*

Haching rate (%, 72 hpf, 3 biological replicates)

94.5 ± 2.94

92.1 ± 6

89.4 ± 3.29

F1 body length (mm, 120 hpf, 20 individuals)

3.91 ± 0.02

3.88 ± 0.02

3.92 ± 0.02

Displacement distance (mm, 0 – 10 min, 120 hpf, 20 individuals)

582.7 ± 106.2

678.4 ± 109

465.4 ± 58.6

Statistical significancedefined as p < 0.05, indicated by an asterisk.

Reviewer 2 Report

In this manuscript authors reviewed the recent studies on the assessment of antimicrobial toxicity on zebrafish. Although relevant studies were summarized in the article, overall, authors need to use exact word, improve the clarity, and writer the manuscript succinctly. e.g., The first sentence is not clear - Antibiotic can inhibit and or kill the growth of non-harmful microorganisms too.

Since toxicity highly depends on concentration, authors need to provide specific information on the concentration of antibiotic in the water as a contaminant, which can help readers to understand which toxicity studies are relevant in terms of concentration that has been used.

Authors need to provide a graphical abstract to provide an overview of the review.

When required authors also need to incorporate toxicity data from other fish, mouse, or other models, and need to compare the data to extract the emerging characteristic(s) from a set of studies.

In line 138, authors need to mention anti-microbial resistance. Authors need to check manuscript carefully to correct similar mistakes.

Author Response

Reviewer 2

In this manuscript authors reviewed the recent studies on the assessment of antimicrobial toxicity on zebrafish. Although relevant studies were summarized in the article, overall, authors need to use exact word, improve the clarity, and writer the manuscript succinctly. e.g.,

1.The first sentence is not clear - Antibiotic can inhibit and or kill the growth of non-harmful microorganisms too.

 Our response: We apologize for the confusion and we agree with the reviewer that more accurate information would be useful for the reader. We have revised for clarity.

Changes in the manuscript:

(Page 2) Antibiotics are natural, synthetic or semi-synthetic compounds which are able to kill or inhibit the growth and metabolic activity of microorganisms

  1. Since toxicity highly depends on concentration, authors need to provide specific information on the concentration of antibiotic in the water as a contaminant, which can help readers to understand which toxicity studies are relevant in terms of concentration that has been used.

 Our response: As suggested by the reviewer, we have added the requested information.

Changes in the manuscript:

(Page 3) For example, monitoring study has shown that sulfonamides have been frequently detected in surface waters and wastewaters with concentrations ranging from 8.4 ng/L to 211 mg/L [45]. Erythromycin concentrations in surface water bodies were found ranging from 0.1 ng/L to 1 mg/L [29]. In Tai Lake (China) the maximum concentrations increased in the order of macrolides (48.8 ng/L) < quinolones (210.67 ng/L) < b-lactams (361.74 ng/L) < tetracyclines (551.18 ng/L) < amphenicols (2.7 mg/L) < sulfonamides (10.2 mg/L) [46]. Additionally, high levels of norfloxacin (up to 2.8 mg/L), ofloxacin (up to 2.4 mg/L), azithromycin (up to 1.7 mg/L) has been reported in influent of 14 municipal wastewater treatment plants [47]. As expected, hospitals were a significant source of antibiotics, the sulfamethoxazole (0.4- 2.1 mg/L), trimethoprim (2.9 – 5 mg/L), ofloxacin (25.5 – 35.5 mg/L), ciprofloxacin (0.85 – 2 mg/L), lincomycin (0.3 – 2 mg/L) and penicillin G (0.85 – 5.2 mg/L) were measured in hospital effluent in New Mexico [48]. Another study performed in Coimbra (Portugal) reported high concentrations of ofloxacin (0.3 – 10.6 mg/L) and ciprofloxacin (0.1 – 11 mg/L) in hospital effluents [49]. Whereas, the concentration of enrofloxacin in the swine wastewater were up to 1.793 mg/L and levels > 0.1 mg/L were observed for sulfamonomethoxine, trimethoprim, ciprofloxacin, ofloxacin, lincomycin, tetracycline from certain pig farms [50]. The removal percentage of oxytetracycline in the wastewater treatment was 38 ± 10.5 % and the antibiotic concentration was still as high as 19.5 ± 2.9 mg/L in the treated outflow [51]. In addition, the concentration slightly decreased along the river to 377 ± 142 mg/L at a site ~20 km away from the discharging point. The maximum concentrations of tetracycline, oxytetracycline and doxytetracycline were found in the dry season as 11.16, 18.98 and 56.09 ng/L, respectively through a survey of water sources along the lower Yangtze River (China) [52]. In summary, the concentration and distribution of antibiotics in aquatic environment are influenced by climatic properties, as well by the pollution source and physico-chemical characteristics of antibiotics.

  1. Authors need to provide a graphical abstract to provide an overview of the review.

Changes in the manuscript:

  1. When required authors also need to incorporate toxicity data from other fish, mouse, or other models, and need to compare the data to extract the emerging characteristic(s) from a set of studies.

 Our response: As suggested by the reviewer, we have added the requested information.

Changes in the manuscript:

(Page 4) The main concerns related to the use and presence of the antibiotics in the environment are anti-microbial resistance and the toxicity to the organisms. The results revealed that antibiotics may have a negative impact on organisms in the aquatic ecosystems, including freshwater algae, microphytes, macrophytes, zooplankton and fishes [57-61]. Commonly used fish for toxicity testing are rainbow trout (Oncorhynchus mykiss) [62, 63], common carp (Cyprinus carpio) [45, 64, 65], goldfish (Carassius auratus) [66], medaka (Oryzias latipes) [60, 61, 67-69], gilthead seabream (Sparus aurata) [70-72], Tra catfish (Pangasianodon hypophthalmus) [73] or zebrafish (Danio rerio). For example, acute (0.01 – 10 mg/L) and chronic exposure (0.05 – 0.8 mg/L) to erythromycin caused histological changes in gills and liver of rainbow trout [62]. In addition, oxidative effects and genotoxicity have been reported, following the exposure of rainbow trout to oxytetracycline [63]. Many changes were found in hematological profile and biochemical parameters of Cyprinus carpio, upon chronic exposure to sulfamethoxazole (25 – 200 mg/L) [65]. The activity of brain acetylcholine esterase in male goldfish was significantly inhibited by norfloxacin (³ 0.4 mg/L) and the mixture of sulfamethoxazole and norfloxacin (³ 0.24 mg/L), after 4 days of exposure [66]. Also, the activities of 7-ethoxyresorufin O-deethylase, glutathione S-transferase and superoxide dismutase in liver and serum vitellogenin were increased by norfloxacin and mixtures. Treatment with various doses of erythromycin ranging from 2 mg/L to 2 mg/L for 96 h decreased the burst speed and swimming speed in medaka [68]. A 10 days treatment with sulfadiazine and trimethoprim in gilthead seabream juveniles induced significant changes in the expression of 41 liver proteins [74]. Additionally, enrofloxacin has been demonstrated to induce e lipid peroxidation and neural dysfunction on Tra catfish [73]. In recent years, many studies have focused on the toxic effects of single antibiotics on embryos, larvae or adults zebrafish. Developmental toxicity, oxidative stress, nephrotoxicity, cardiotoxicity, oculotoxicity, neurotoxicity or ototocixity (Table 1) have been observed, following both acute and chronic exposure to antibiotics.

  1. In line 138, authors need to mention anti-microbial resistance. Authors need to check manuscript carefully to correct similar mistakes.

Our response: As suggested by the reviewer, we have revised the sentence for clarity.

Changes in the manuscript:

(Page 4) The main concerns related to the use and presence of the antibiotics in the environment are anti-microbial resistance and the toxicity to the organisms.

Reviewer 3 Report

Overall, this is a clear, succinct, and well-written review article. Adequate information about the previous findings has been presented for the readers to follow up the article. The abstract provides a concise and complete summary. The reference list is appropriate, but author authors can certainly incorporate more relevant papers to the reference list. I recommend this article for publication.

Author Response

Reviewer 3

Overall, this is a clear, succinct, and well-written review article. Adequate information about the previous findings has been presented for the readers to follow up the article. The abstract provides a concise and complete summary. The reference list is appropriate, but author authors can certainly incorporate more relevant papers to the reference list. I recommend this article for publication.

Our response: As suggested by the reviewer, we have added the requested information; more relevant papers have been added to the reference list.

(Page 3) For example, monitoring study has shown that sulfonamides have been frequently detected in surface waters and wastewaters with concentrations ranging from 8.4 ng/L to 211 mg/L [45]. Erythromycin concentrations in surface water bodies were found ranging from 0.1 ng/L to 1 mg/L [29]. In Tai Lake (China) the maximum concentrations increased in the order of macrolides (48.8 ng/L) < quinolones (210.67 ng/L) < b-lactams (361.74 ng/L) < tetracyclines (551.18 ng/L) < amphenicols (2.7 mg/L) < sulfonamides (10.2 mg/L) [46]. Additionally, high levels of norfloxacin (up to 2.8 mg/L), ofloxacin (up to 2.4 mg/L), azithromycin (up to 1.7 mg/L) has been reported in influent of 14 municipal wastewater treatment plants [47]. As expected, hospitals were a significant source of antibiotics, the sulfamethoxazole (0.4- 2.1 mg/L), trimethoprim (2.9 – 5 mg/L), ofloxacin (25.5 – 35.5 mg/L), ciprofloxacin (0.85 – 2 mg/L), lincomycin (0.3 – 2 mg/L) and penicillin G (0.85 – 5.2 mg/L) were measured in hospital effluent in New Mexico [48]. Another study performed in Coimbra (Portugal) reported high concentrations of ofloxacin (0.3 – 10.6 mg/L) and ciprofloxacin (0.1 – 11 mg/L) in hospital effluents [49]. Whereas, the concentration of enrofloxacin in the swine wastewater were up to 1.793 mg/L and levels > 0.1 mg/L were observed for sulfamonomethoxine, trimethoprim, ciprofloxacin, ofloxacin, lincomycin, tetracycline from certain pig farms [50]. The removal percentage of oxytetracycline in the wastewater treatment was 38 ± 10.5 % and the antibiotic concentration was still as high as 19.5 ± 2.9 mg/L in the treated outflow [51]. In addition, the concentration slightly decreased along the river to 377 ± 142 mg/L at a site ~20 km away from the discharging point. The maximum concentrations of tetracycline, oxytetracycline and doxytetracycline were found in the dry season as 11.16, 18.98 and 56.09 ng/L, respectively through a survey of water sources along the lower Yangtze River (China) [52]. In summary, the concentration and distribution of antibiotics in aquatic environment are influenced by climatic properties, as well by the pollution source and physico-chemical characteristics of antibiotics.

(Page 4) The main concerns related to the use and presence of the antibiotics in the environment are anti-microbial resistance and the toxicity to the organisms. The results revealed that antibiotics may have a negative impact on organisms in the aquatic ecosystems, including freshwater algae, microphytes, macrophytes, zooplankton and fishes [57-61]. Commonly used fish for toxicity testing are rainbow trout (Oncorhynchus mykiss) [62, 63], common carp (Cyprinus carpio) [45, 64, 65], goldfish (Carassius auratus) [66], medaka (Oryzias latipes) [60, 61, 67-69], gilthead seabream (Sparus aurata) [70-72], Tra catfish (Pangasianodon hypophthalmus) [73] or zebrafish (Danio rerio). For example, acute (0.01 – 10 mg/L) and chronic exposure (0.05 – 0.8 mg/L) to erythromycin caused histological changes in gills and liver of rainbow trout [62]. In addition, oxidative effects and genotoxicity have been reported, following the exposure of rainbow trout to oxytetracycline [63]. Many changes were found in hematological profile and biochemical parameters of Cyprinus carpio, upon chronic exposure to sulfamethoxazole (25 – 200 mg/L) [65]. The activity of brain acetylcholine esterase in male goldfish was significantly inhibited by norfloxacin (³ 0.4 mg/L) and the mixture of sulfamethoxazole and norfloxacin (³ 0.24 mg/L), after 4 days of exposure [66]. Also, the activities of 7-ethoxyresorufin O-deethylase, glutathione S-transferase and superoxide dismutase in liver and serum vitellogenin were increased by norfloxacin and mixtures. Treatment with various doses of erythromycin ranging from 2 mg/L to 2 mg/L for 96 h decreased the burst speed and swimming speed in medaka [68]. A 10 days treatment with sulfadiazine and trimethoprim in gilthead seabream juveniles induced significant changes in the expression of 41 liver proteins [74]. Additionally, enrofloxacin has been demonstrated to induce e lipid peroxidation and neural dysfunction on Tra catfish [73]. In recent years, many studies have focused on the toxic effects of single antibiotics on embryos, larvae or adults zebrafish. Developmental toxicity, oxidative stress, nephrotoxicity, cardiotoxicity, oculotoxicity, neurotoxicity or ototocixity (Table 1) have been observed, following both acute and chronic exposure to antibiotics.
